

# Visible implant elastomer (VIE) success in early larval stages of a tropical amphibian species

Chloe A. Fouilloux, Guillermo Garcia-Costoya and Bibiana Rojas

Department of Biology and Environmental Science, University of Jyväskylä, Jyväskylä, Finland

## ABSTRACT

Animals are often difficult to distinguish at an individual level, and being able to identify individuals can be crucial in ecological or behavioral studies. In response to this challenge, biologists have developed a range of marking (tattoos, brands, toe-clips) and tagging (banding, collars, PIT, VIA, VIE) methods to identify individuals and cohorts. Animals with complex life cycles are notoriously hard to mark because of the distortion or loss of the tag across metamorphosis. In amphibians, few studies have attempted larval tagging and none have been conducted on a tropical species. Here, we present the first successful account of VIE tagging in early larval stages (Gosner stage 25) of the dyeing poison frog (*Dendrobates tinctorius*) coupled with a novel anesthetic (2-PHE) application for tadpoles that does not require buffering. Mean weight of individuals at time of tagging was 0.12 g, which is the smallest and developmentally youngest anuran larvae tagged to date. We report 81% tag detection over the first month of development, as well as the persistence of tags across metamorphosis in this species. Cumulative tag retention vs tag observation differed by approximately 15% across larval development demonstrating that "lost" tags can be found later in development. Tagging had no effect on tadpole growth rate or survival. Successful application of VIE tags on *D. tinctorius* tadpoles introduces a new method that can be applied to better understand early life development and dispersal in various tropical species.

# INTRODUCTION

Distinguishing individuals within a population is often key in deciphering animal behavior, life-history traits, and ecological dynamics. Animal identification has applications in understanding parental care (*Ménard et al., 2001*), migration dynamics (*Matthews et al., 2011*; *Fuller et al., 2008*), adaptations to environmental pressures (*Warne & Crespi, 2015*; *Gordon et al., 2009*), and even fecundity (*Martin, 1995*). Studies across the animal kingdom have developed methods that vary both in invasiveness and success (guppies: *Croft et al. (2003)*, *Gordon et al. (2009)*; salamanders: *Osbourn et al. (2011)*; turtles: *Fuller et al. (2008)*; birds: *Martin (1995)*; dolphins: *Defran, Shultz & Weller (1990)*; bears: *Diefenbach & Alt (1998)*) to allow researchers to differentiate between individuals within groups. If visual differentiation is not an obvious option, physical

Corresponding author
Chloe A. Fouilloux, chloe.a.fouilloux@jyu.fi

manipulation (e.g., toe clips, tattoos; *Perret & Joly, 2002*; *Phillott et al., 2007*) and tagging (e.g., passive integrated transponder (PIT), *Perret & Joly (2002)*; visible implant alphanumeric (VIA), *Caballero-Gini et al. (2019)*; visible implant elastomer (VIE), *Brannelly, Chatfield & Richards-Zawacki (2013)*; ear tags, *Diefenbach & Alt (1998)*; banding, *Martin (1995)*; and collars, Gese (2001)) have been the most commonly used methods implemented in mark-recapture studies.

Differentiating individuals is important when there is a lot of intrapopulation variation in behavior, and is becoming especially relevant as we begin to see individuals adapt to new challenges onset by the effects of global warming, habitat fragmentation, and human interactions. However, long-term mark-based studies have not often been applied across ontogenetic stages of animals with complex life cycles, as the physical transformations induced with metamorphosis and growth generally entail the loss or distortion of the mark. In amphibians, there have been a range of successful tagging methods both in adult and larval stages, but the diversity of larval tagging studies has been limited to common temperate species (*Campbell Grant, 2008*; *Courtois et al., 2013*), and very few studies have been able to create a methodology that spans the animal's entire life cycle (*Bailey, 2004*; *Bainbridge et al., 2015*; *Caballero-Gini et al., 2019*; *Campbell Grant, 2008*; *McHarry et al., 2018*).

Understanding the dispersion dynamics and survival of amphibians from aquatic to terrestrial habitats makes developmentally early larval tagging especially interesting. Tagging in amphibian tadpoles could be used to understand how environmental stress affects individual development in group conditions (e.g., *Dendropsophus ebracattus* (*Touchon, Urbina & Warkentin, 2011*), *Agalychnis callidryas* (*Gonzalez, Touchon & Vonesh, 2011*), *Triturus alpestris* (*Denoel & Joly, 2000*)) or could be used to investigate the dynamics of aggressive displays interactions between tadpoles (e.g., *Ambystoma tigrinum nebulosum* (*Pfennig, Loeb & Collins, 1991*), *Rana utricularia* (*Faragher & Jaeger, 1998*), *Oophaga pumilio* (*Dugas, Stynoski & Strickler, 2016*)). Many species of Neotropical poison frogs have parental care where recently hatched tadpoles are transported from terrestrial sites to arboreal pools (*Pašukonis, Loretto & Rojas, 2019*; *Ringler et al., 2013*; *Schulte & Mayer, 2017*; *Summers & Tumulty, 2013*). Tadpole tagging could provide a quick and reliable method of following individuals across development, understanding relatedness within pools, and observing tadpole behavior and parental care in the field. In this study, we mark the larvae of *Dendrobates tinctorius*, a neotropical species of poison frog whose tadpoles develop in ephemeral pools of water.

Larval anuran tagging has been limited with respect to both developmental stage and weight. Most of the inter-stage larval tagging to date has been done beyond the point of the onset of hind leg development (*Andis, 2018* (Gosner stage 30); *Bainbridge et al., 2015* (Gosner stage 36–38); *Gosner, 1960*). At this stage, *D. tinctorius* tadpoles are typically at least a month old, meaning they have already been transported by their fathers and have long since been subject to both predation risk and aggression by conspecifics (*Rojas, 2014*, *2015*; *Rojas & Pašukonis, 2019*). Therefore, in order to obtain more valuable life history information, tags need to be applied earlier in development.

To our knowledge, the developmentally earliest tagging study applied VIA/VIE tags around Gosner stage 25 (*Courtois et al., 2013*), but its application was limited to large temperate tadpoles (average weight around 1.5 g) that could be manipulated in the field without anesthesia. In this study we use 2-phenoxyethanol (2-PHE), an anesthetic that does not need to be buffered and can be stored at room temperature, making it field appropriate (*Acme-Hardestry, 2013*; *National Center for Biotechnology Information, 2020*). 2-PHE has been used on newts (*Perret & Joly, 2002*), fishes (*Toni et al., 2015*), and adult frogs (*Eggert, Peyret & Guyétant, 1999*) for anesthetic purposes, but has been largely overlooked for larval application. In most amphibian research MS-222 has been used, popularized perhaps because of its use in amphibian medicine (*Mitchell, 2009*; *Vrskova & Modra, 2012*). Yet, MS-222 needs to be buffered and has been found to increase cortisol concentrations (an indicator of stress) (*Coyle, Durborow & Tidwell, 2004*). In comparison, studies using 2-PHE on fish found that this anesthetic prevented the induction of stress pathways during stressful procedures (*Toni et al., 2015*). Further, 2-PHE has a large safety margin for applied doses and can be easily acquired through common compound manufacturers (*Matthews & Varga, 2012*).

In this study, we apply VIE tags to the smallest and developmentally earliest stages of larval anurans recorded to date. We follow growth rate and tag success across larval development, and discuss potential field applications in order to better understand the dispersion dynamics and behavior of protected frogs.

## MATERIALS AND METHODS

### Study organism

*Dendrobates tinctorius* is a neotropical poison frog that transports their recently hatched larvae to ephemeral pools of water. In addition to the risk of desiccation, tadpoles face predation by their cannibalistic conspecifics, as well from heterospecifics (e.g., Odonate naiads) that occur in these pools (*Rojas, 2014*, *2015*). The larval period lasts approximately 2 months in the wild (*Rojas & Pašukonis, 2019*) though the laboratory population has had a longer range (2.5–3 months).We used tadpoles from a breeding laboratory population of *Dendrobates tinctorius* kept at the University of Jyväskylä, Finland. Adult pairs were each housed in a 55L terrarium that contained layered gravel, leaf-litter, moss substrate and was equipped with a shelter, ramps, and live plants. Terraria were maintained at 26 °C (±2 °C) and were automatically misted with reverse osmosis water four times a day, maintaining a humidity around 95%. They were lit with a 12:12 photoperiod. Frogs were fed live *Drosophila* fruit flies coated in vitamin supplements three times per week. Tadpoles were raised singly in 10 × 6.5 × 5 cm cups which were filled with spring water, and fed an ad libitum diet of fish food (JBL NovoVert flakes) three times a week. Adult and tadpole health and water levels were checked daily, and experimental tadpoles were weighed and photographed weekly. Experiments began in October 2019 and continued through April 2020. This experiment was permitted by the National Animal Experiment Board (ESAVI/9114/04.10.07/2014).

## Tags

Visible implant elastomers are a 2-part silicone-based polymer that is injected as a liquid that hardens to a pliable consistency once warmed (VIE; Northwest Marine Technology Inc., Anacortes, WA, USA). The result is a small color band on the surface of the animal that can be detected by the naked eye. There is a range of 10 possible colors for application, 6 of which are fluorescent. Visualization of fluorescent tags can be enhanced using a UV light. VIE tags have been successfully used in diverse taxa across developmental stages (e.g., echinoderms: *Martinez, Byrne & Coleman (2013)*; fish: *Croft et al. (2003)*; salamanders: *Campbell Grant (2008)*).

## Anesthesia

Prior to tagging, tadpoles were anesthetized in a 14 mL solution of a one µl:one mL ratio of 2-PHE to spring water. 2-PHE is an oily liquid at room temperature and does not need to be buffered for anesthetic purposes (*Coyle, Durborow & Tidwell, 2004*). The solution was reused multiple times for multiple tadpoles within a single day of tagging (max. 10 tadpoles tagged each session); its effect did not deteriorate after multiple uses. Each day of tagging a new solution was made. Tadpoles were placed in an anesthetic solution until there was no muscular contraction in response to agitation; this process took approximately 3 min. We assumed the anesthetic's potency did not degrade because the latency of its effects remained consistent after being applied to multiple tadpoles. The effect of anesthesia on tadpoles lasted approximately 6 min; within 10 min individuals had regained full muscular function. The effects of anesthesia were similar across developmental stages (Gosner 24–26). We had no deaths in response to our anesthesia procedure which was applied to a total of 40 individuals across both our pilot study and experimental manipulations.

## Tadpole tagging

We applied VIE tags to early larval stages of *D. tinctorius* and monitored tadpoles across development (Fig. 1) to ensure the presence of the tags over time, and to test the effects of larval tagging and tag retention. Previous studies reporting tadpole tagging have been done primarily with late-term tadpoles (Gosner stage 30+) whose snout-vent lengths (SVL) were double or triple the SVL of tadpoles in our experiment (*Andis, 2018*; *Bainbridge et al., 2015*; *McHarry et al., 2018*). Other studies also worked with amphibians who produce large egg clutches (*Litoria aurea*, 37,000 eggs/clutch (*Pyke & White, 2001*); American bullfrog, 12,000 eggs/clutch (*Howard, 1978*); *Alytes obstetricans*, 50 eggs/clutch (*Reading & Clarke, 1988*)), which allowed for large tag sample sizes ($n = 53–90$, depending on study). *Dendrobates tinctorius* lay clutches that range from 2 to 5 eggs with a high level of mortality (*Rojas & Pašukonis, 2019*). Due to the reproductive limitations of the system, our sample total ($n = 27$ tagged, $n = 11$ control) is less than previously published data.

Elastomer was mixed and loaded into syringes prior to each tagging session, according to the Northwest Marine Technology VIE tag protocol. Elastomer was stored in a freezer (−20 °C) during extended periods of disuse and in a refrigerator between individual tagging sessions; we found that mixed elastomer was no longer applicable after a storage

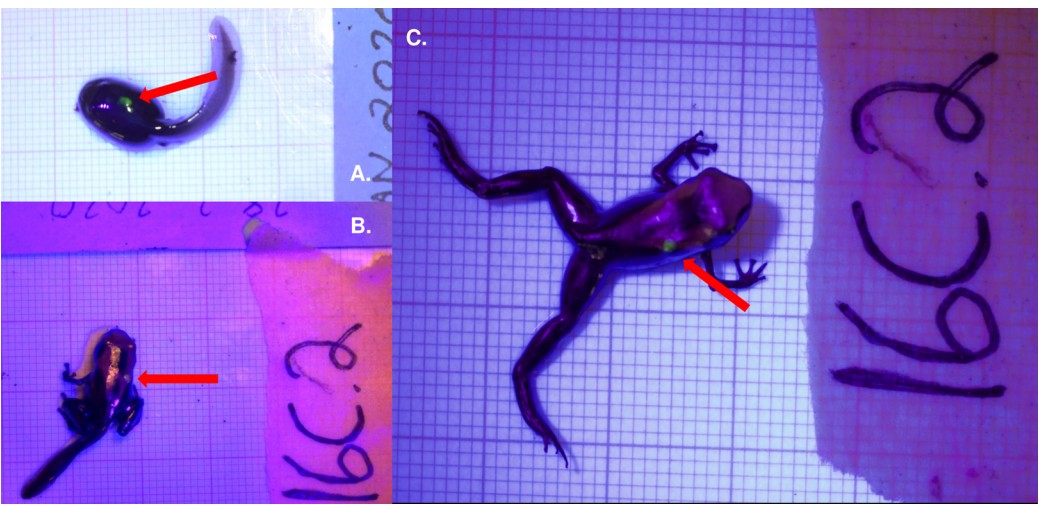

**Figure 1 Fluorescent green VIE tag inserted dorsally on *Dendrobates tinctorius*.** Tag shown on the same individual as (A) a late stage larva, (B) a metamorph, and (C) a recently metamorphosed juvenile. All photos taken with Nikon DS5300 DSLR on 1 × 1 mm background under UV light to enhance tag detection.

period of longer than 3 months. Average tagging procedure was executed in under 90 s. Throughout our pilot study we found that tag retention was most effective when placed dorsally; thus, this experiment only contained dorsally marked tadpoles. Each tadpole was marked only once.

Immediately after being anesthetized, tadpoles were prepared for tagging. This was done by removing tadpoles from the anesthetic solution and placing them on a laminated surface where they were dried with a paper towel to improve grip. Once excess moisture was removed from the body (without completely drying out the tadpole), a three mL insulin syringe with a 30 G/12 7 mm needle was placed subcutaneously and dye (approx. one μl) was injected. For this experiment, we used a fluorescent green elastomer, though any color tag would have been suitable for application. After tag injection, tadpoles were placed under UV light to ensure proper placement of the tag. Proper placement was qualified as the tag being injected deep enough to not fall out (directly under epidermis) but shallow enough to be visible with the help of a UV light. Tadpoles were then cleaned with spring water and the status of the tag was checked again. Tadpoles post-tagging were placed in a pool of spring water and observed for 10 min to ensure proper return of muscular function. After the observation period, tadpoles were returned to the pool of water in which they were living.

## Tag observation

Tags were observed once a week by a single person. Tadpoles were placed on a laminated surface, cleaned with spring water, dried, and checked both dorsally and ventrally for the presence of a tag using a UV light. Observers were not blind to the treatment or identity of each tadpole, as the observer both photographed and weighed all experimental tadpoles within a recording session.

## Statistical analysis

### Tag retention and observation model

Visible implant elastomer tag retention and observation was modeled using a Bayesian Cormack–Jolly–Seber (CJS) survival model (see R and JAGS code in Supplemental Materials; *Jolly, 1965*; *Lebreton et al., 1992*; *Seber, 1965*). For each individual, we considered tag observation as a categorical variable that was recorded as absent (0) or present (1); tags that had been lost and not re-observed were marked NA after the last confirmed observation. We assessed the status of the tag and tadpole development (size, weight) weekly. Tag retention was recorded as present (1) for all weeks previous to the last observation and recorded as NA for all those that followed. Our coding schematic takes into account observer error as a tag that is not observed at one time point but seen later in development is recorded as "retained" throughout the entire unobserved period. The retention status of the tag is unknown after the last positive observation. Distinguishing observation and retention rates allowed us to calculate the rate of false negatives in tag observation. Our model considered weekly discrete time steps where the retention and observation of the tag were considered latent variables that occurred with a certain probability ($\phi$ and $p$, respectively following the nomenclature commonly used for CJS models).

We considered five possible models (M1–5) of increasing complexity for $\phi$ and $p$: M1 assumed constant probabilities of observation and retention, M2 considered a week effect on both probabilities using a logit link function, M3 took into account both a week effect on $\phi$ and $p$ as well as individual identity as a random effect, M4 had the same parameters as M3, but considered the weight at time of tagging, and M5 had to same parameters of M4 but included individual identity as a random effect. Tag retention and observation were defined as following a binomial distribution with a probability $\phi$ and $p$ respectively for all models. In models M2–M5 retention and observation varied for each week of development ($t$), thus we used a logit link function to determine $\phi$ and $p$ for each week considered. In model M3 and M5 retention and observation were also influenced by individual identity (*id*), to account for it, we sampled the random effect parameter estimates from a normal distribution with a certain standard deviation for each individual which were later incorporated to the same logit link function.

For each model we used an MCMC approach considering uninformative priors for all parameters (see Supplemental Materials) and simulation run characteristics of 4 chains, 100,000 iterations with a 5,000 burn-in and a thinning of 10. Chain convergence was assessed using a potential scale reduction factor (PSRF) of our parameter estimates which discarded any model run that resulted in a PSRF larger than 1.1 or smaller than 0.9. We checked sample independence by determining the effective sample size of each parameter. We did not consider any model run with less than 5,000 independent samples for any parameter.

Model selection was based on the lowest DIC value (Deviance Information Criterion) and biological relevance. The most likely discrete probabilities of retention and observation for each week were based on the posterior distributions generated by our model, these

values were used to visualize the cumulative probability of tag detection across larval development.

### Tadpole growth rates

Growth rates were compared between treatments using a linear mixed-effect model (LMM). Weekly weight (~weight) and treatment (~treatment) were coded as additive predictors in the growth rate model. Tadpole ID was used as a random effect on the intercept. Growth between treatments was compared by calculating weekly rate changes across development for both treatments. Rates percent were used in model analysis which were then evaluated with a Kenward-Roger's method ANOVA. Growth rate models were chosen as a result of Akaike Information Criterion output (AIC; *Akaike, 1973*).

### Tadpole survival rates

We used a Kaplan–Meier survival curve to visualize treatment effect on tadpole survival. A mixed effects Cox model was used to calculate the parameters and uncertainty of tagging on survival. Survival object was parameterized with respect to death and time in response to treatment and took individual tadpole identity into account as a random effect (Surv(Week, Dead) ~ Treatment + (1|ID)). Survival was coded as a binomial response (alive (0), dead (1)). We took a frequentist approach in modeling tadpoles survival rates as it has been suggested that in the absence of reliable informative priors, Cox models are preferable for survival data (*Omurlu, Ozdamar & Ture, 2009*).

All models and statistics were performed in the program R using base R (v. 3.6.1, *R Core Team, 2019*) with additional packages "survival" (*Therneau, 2014*), "coxme" (*Therneau, 2020*), "dplyr" (*Wickham et al., 2019*), "lme4" (*Bates et al., 2015*), "pbkrtest" (*Halekoh & Højsgaard, 2014*), "JAGS" (*Plummer, 2003*), and "R2jags" (*Su & Yajima, 2015*).

## RESULTS

### Tag success

Out of our 27 fluorescent tags, 81% (22/27) were successfully detected in tadpoles over the first month of application. This decreased to a little over 50% (8/15) detection by the third month of application, which also marks the approximate time of tadpole metamorphosis. Tags were observed in four out of the 11 tadpoles (36%) that survived past metamorphosis. None of our experimental frogs retained their tags after 5 months of development. Mean weight at time of tagging was 0.12 g (±0.019 SE) for tagged tadpoles and 0.099 g (±0.015 SE) for control tadpoles. Control tadpole weights ranged from 0.0307 to 0.18 g at initial weigh-in, tagged tadpole weights ranged from 0.0318 to 0.36 g at time of tagging. The smallest successful tag was applied at 0.0318 g, which was a tadpole who had recently hatched (approximately Gosner stage 25). Our experimental tadpoles were tagged in the early larval stages of development: the youngest successful tag was applied on recently hatched tadpoles who had yet to be transported by their fathers. Tagging did not seem to prevent transport behavior by the father, although we observed transport of tagged tadpoles in only two instances. Tagging at this life stage is especially

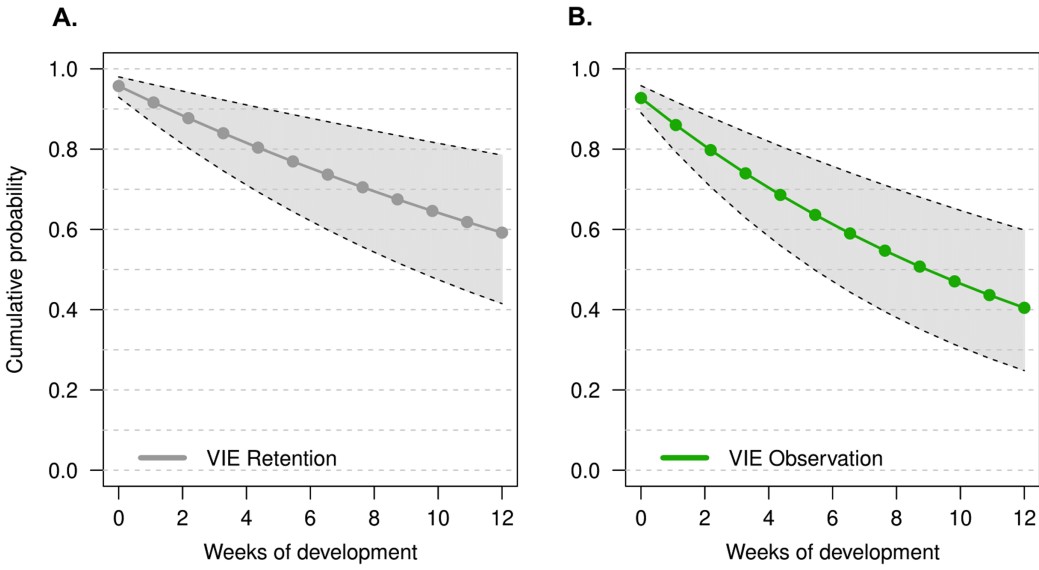

**Figure 2 Estimate of the cumulative probability of tag detection across larval development for model M1.** (A) Probability of tag retention across larval development. Grey points are the probability of tag retention ($\phi$). (B) Probability of tag observation across larval development. Green points are the probability of tag observation ($p$). All points are posterior means at discrete time intervals corresponding to weeks of development. Grey polygons delimited by black dashed lines indicate the 95% credible intervals.                

delicate and requires a practiced hand. We attempted embryonic tagging in pilot studies, but were not able to successfully inject the tag without permanently damaging the embryo.

The model that had the lowest DIC did not include a week effect or an individual random effect (M1). There was no improvement in model quality when including tadpole mass at time of tagging, suggesting that initial tadpole mass had no effect on tag observation throughout development. It is important to note that tag observation sometimes changed throughout development, and tags that were not observed 1 week sometimes were detectable later in development (see Fig. 2). Instances where tags were not observed could be due to individual growth, resulting in a tag being obstructed by a physical structure (i.e., muscle, tissue) for a period of time. For example, the cumulative probability of tag retention ($\phi$) until the third month of development was 0.61 (95% CI [0.33–0.80]) while the cumulative probability of tag observation ($p$) was 0.38 (95% CI [0.16–0.59]), this demonstrates that after 12 weeks the rate of false negatives is approximately 23%. On average, the difference between cumulative retention and observation rates was about 15%.

## Growth rate

We found no significant difference in weekly growth rate between control and tagged tadpoles (Fig. 3), indicating that tagging does not affect tadpole growth (lmer, ANOVA Kenward–Roger's method, $F(1, 37) = 1.12$, $p = 0.296$). Weekly tadpole growth rate significantly decreased across time (lmer, ANOVA Kenward–Roger's method, $F(1, 415) = 56.4$, $p = 0.03563^{-11}$).

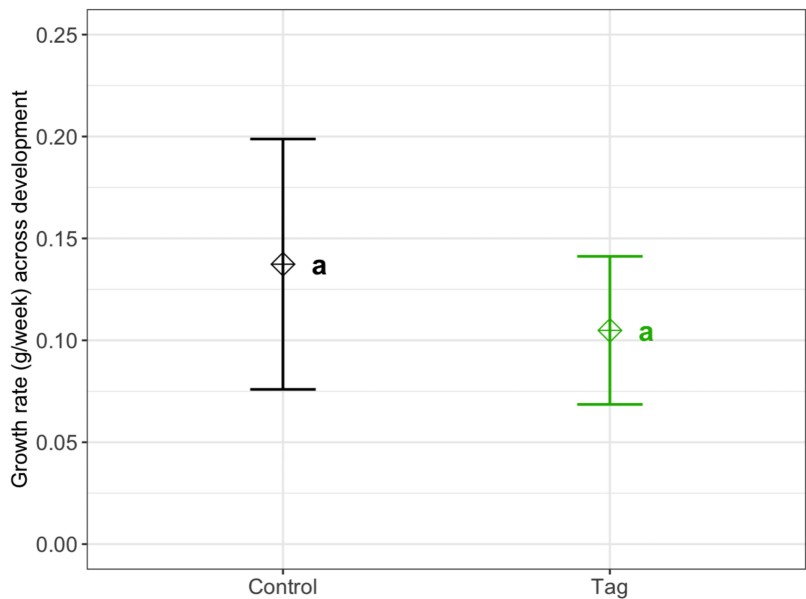

**Figure 3 Average growth rate of VIE tagged and control group tadpoles.** Diamonds represent the LS mean for which error bars indicate 95% confidence intervals. Means sharing letters are not significantly different (Tukey-adjusted comparisons, ANOVA Kenward–Roger's method, $F(1, 37.97) = 1.12$, $p = 0.296$).

## Survival

There was no significant difference in survival between control and tagged groups across larval stages of development. Mortality across the first 3 months was 18% ($n = 5/27$) for tagged tadpoles and 27% ($n = 3/11$) for control tadpoles (Fig. 4). A mixed effects Cox model did not find any significant difference in survival based on treatment (coxme, $z = 0.09$, $p = 0.93$). Post-metamorphic survival was excluded from analysis due to unnaturally high froglet loss throughout the lab colony which is not indicative of tag impact on froglet survival, but likely ineffective laboratory practices for juvenile health. At time of publication (May 2020) $n = 3$ tagged tadpoles and $n = 1$ control tadpoles were alive.

## DISCUSSION

In our study we applied VIE tags on *D. tinctorius* tadpoles and monitored them across larval development under laboratory conditions. Compared to previously published visible implant studies, our approach presents application at the youngest developmental stage, and is one of the first studies (after *Bainbridge et al., 2015*; *Warne & Crespi, 2015*; *Andis, 2018*) to follow tags across metamorphosis. To our knowledge, this is the first attempt of larval tagging in a tropical frog, as previous work focused exclusively on species from temperate regions (*Bainbridge et al., 2015*: *Litoria aurea*; *Courtois et al., 2013*, *Alytes obstetricans*; *Nauwelaerts, Coeck & Aerts, 2000*, *Rana esculenta*). The successful application of VIE tags for the first time in a tropical species with elaborate parental care provides valuable opportunities to investigate parent-offspring interactions and dispersion of these species in a natural context.

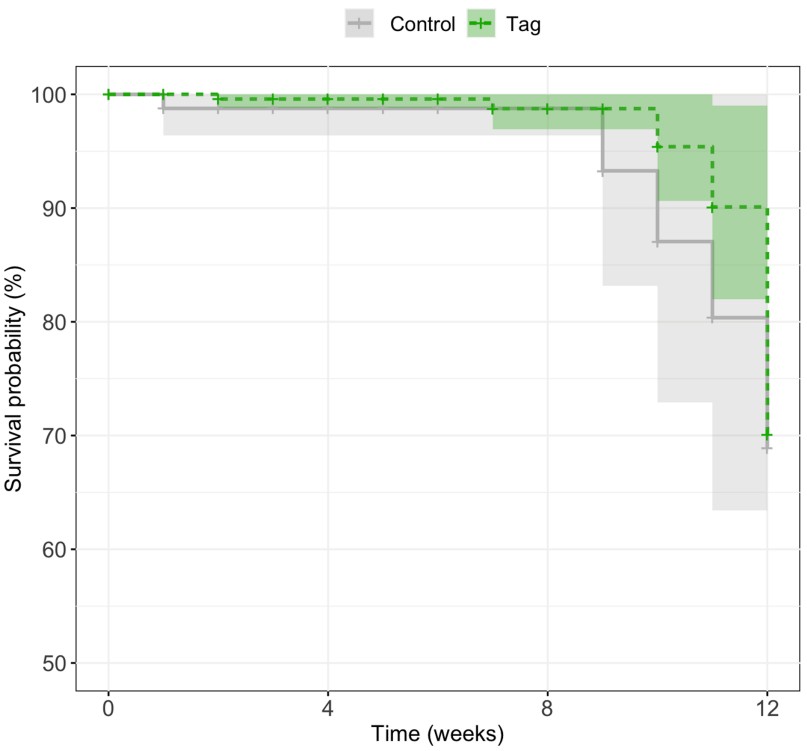

**Figure 4 Kaplan–Meier survival curves of experimental tadpoles across larval development.** Curves represent the probability of survival of tagged and control tadpoles over time where the light green line represents tagged tadpoles and the light grey bar represents control tadpoles. A mixed effects Cox model did not find any significant difference of treatment on tadpole survival across development ($z = 0.09$, $p = 0.93$). Shaded regions represent 95% confidence intervals.

Similar to other studies, we found no difference in growth rate or survival between tagged and control treatments. Based on our weekly weigh-ins and LMM model, we did not detect any significant impact of tagging on growth rates across development (see Fig. 3). Throughout our study, tadpoles grew significantly faster earlier in development which could be due to laboratory conditions (a high-food, no-competition environment). Given the circumstances, tadpoles may have invested energy in growing earlier in development which would help avoid predation and decrease latency to metamorphosis in the wild (*Caldwell & De Araújo, 1998*; *Rojas, 2014*). We found no effect of tagging on *D. tinctorius* survival across development; however, we had high rates of post-metamorphic mortality across our laboratory population which impacted our ability to assess tag success across development. Natural history studies of *D. tinctorius* have shown high larval death rates (*Rojas & Pašukonis, 2019*). Although our tadpoles were not subject to the same pressures as wild populations, the mortality we observed across both treatments reflect the precariousness of early life stages in *D. tinctorius*.

After the first month of observation, 81% of tadpoles retained their tag, which is on par with retention rates reported in other tagging efforts (*Anholt, Negovetic & Som, 1998*; *Martin, 2011*). Other studies report even higher rates of success with tadpole elastomer tags (*Courtois et al., 2013* (100%), *Bainbridge et al., 2015* (100%)) which could be due to a

shorter larval period and larger individuals at time of application. Retention rates of this study are of note because VIE tags have been extensively used for mark-recapture studies in fishes and anurans; when taking into account our tag retention rate and our tagging procedure (which takes less than 90 s), we can conclude that larval VIE mark-recapture studies on tropical amphibians is feasible.

A relevant note about implant tagging is that it is limited to observer perception. As tadpoles develop, morphological and phenotypic individual changes can facilitate or mask the presence of a tag. Most importantly, the lack of tag observation should not be assumed to indicate tag loss. In our experiment we were able to differentiate the cumulative probability of retention vs observation over time as a result of weekly checks of tag condition in experimental tadpoles to account for false negatives. Over 3 months of development tags could go multiple weeks unobserved; finding them later in development indicated that tags were not lost, but had shifted position or been re-exposed as a result of growth. This is important to take into account for mark-recapture studies in settings where regular sampling or capture of the entire tagged population isn't feasible. Our model estimates an average 15% difference in tag observation vs retention across larval development which is an error that can be incorporated as an informative prior in future tagging studies.

Visible implant elastomer tags come in a range of fluorescent colors, making the distinction of clutches or individuals from a distinct cohort possible. This is especially relevant for the larval stages of *D. tinctorius* when tadpoles are aggressive cannibals, as tagging efforts would help distinguish resident tadpoles in phytotelmata. Thus, tagging could be used to help monitor who is being deposited and who is getting attacked, allowing us to track interactions between tadpoles in ephemeral pools (*Rojas, 2015*). Moreover, VIE tagging of *D. tinctorius* makes it possible to successfully tag tadpoles before they are picked up and transported by their parent. Elastomer tags most clearly fluoresce under low-light conditions, making them ideal for their application in wild *D. tinctorius* tadpoles which live in dimly lit closed canopy rainforest.

Visible implant elastomer tags are one of the smallest tagging methods available for field studies. With respect to other tagging methods, VIA tags require a minimum SVL of two cm (*Courtois et al., 2013*) and PIT tags require four cm (*Courtois et al., 2013*), making VIE tags a unique option to study larval dynamics. VIE tags are not more than four mm in length, meaning that their successful application presents new opportunities to study larval amphibians that may not have been considered in the past. For example, *Anomaglossus beebei*, a small endemic poison frog from Guyana, has been seen to transport tadpoles multiple times throughout development (C.A. Fouilloux, 2017, personal observation). Larval tagging of this species could help decipher how shifting male territories influences larval care and transport, and if newly established males take care of tadpoles that are not their own. Early larval tagging could also work for *Allobates femoralis*, another tadpole transporter, to understand the shifting genetic diversity within phytotelmata across time (*Erich et al., 2013*).

Coupled with the unique patterning of *D. tincorius* that emerges in late metamorphosis and settles in adulthood (*Courtois et al., 2012*; *Rojas & Endler, 2013*), tags can provide early

life identification that could be followed by pattern recognition, enabling individual discrimination throughout an individual's entire lifespan. *Bainbridge et al. (2015)* report recently metamorphosed VIE tag retention to be high (88–95%); we also find that tags that lasted throughout larval development persisted across metamorphosis and into terrestrial life. Aside from implant tagging, genetic tracking has proven to be a reliable method to follow amphibian larvae throughout development into adulthood. With this said, genetic tracking does not provide immediate individual detection; further, studies using this method have been limited to individuals in a closed population, making the recapture of (surviving) tracked individuals reasonably certain (*Ringler, Mangione & Ringler, 2015*). In *D. tinctorius*, however, males can travel remarkable distances while carrying tadpoles (*Pašukonis, Loretto & Rojas, 2019*) making genetic tracking a less suitable method for individual distinction in this species. *Andis (2018)* also did important work dyeing tadpoles of *Rana sylvatica* with calcein. This dye appears to persist across metamorphosis, though it should be noted that their development is much shorter than *D. tinctorius* and staining only allows for presence/absence detection. The presence of a VIE tag (and the range of colors available for application) allows for immediate discrimination of multiple groups/cohorts which may be an important advantage when conducting behavioral experiments and elucidating natural history dynamics in the wild.

Our study presents a successful continuation expanding marking methodology to larval tropical species. Using laboratory conditions, we were able to mimic a common scenario where experimental tadpoles were left to develop in small pools of water. This is reflective of the most common parental behavior exhibited by *D. tinctorius*, where males transport newly-hatched tadpoles to develop in small water holdings (*Rojas & Pašukonis, 2019*). Future studies in field conditions would be useful to supplement these findings. For example, it will be important to understand how tadpole interaction with conspecifics, heterospecifics, and predators affects tag retention. It is also important to consider how tagging will affect individual behavior and success. Although different from VIEs, studies have found tadpole staining to have effects both on predator response (*Carlson & Langkilde, 2013*) and aggression levels (*Fischer et al., 2020*); it is important to acknowledge that manipulating animals can entail unexpected/unintended consequences and that further studies working with tags in natural settings are warranted. However, based on previously published data and the observation rates of our elastomers in this experiment, we believe that the application of elastomers in the wild is already an appropriate method to distinguish tadpoles for behavioral experiments. Elastomers are small, successful, and relatively easy to apply in early amphibian life stages. Our study contributes to the growing body of methods-based research demonstrating that visible implant elastomers are a viable tagging solution on a variety of anuran species in early development.

## CONCLUSIONS

Differentiating individuals/cohorts can be a powerful tool when conducting behavioral experiments. Often, marking animals is a technique used to distinguish individuals when

physical features are not distinct enough for visual differentiation. Choosing an optimal tag for a system is a tradeoff between reliability and invasiveness and is often limited to product cost and efficiency in identification. Elastomers (VIE) are injectable polymers that have been extensively used in fish and anuran systems. However, until this point, they have been applied to large larvae or adults and have been heavily biased towards common, temperate species. Here, we present the first application of VIE tags on a small larval tropical frog (*Dendrobates tinctorius*) and follow tag success across development. We found that (1) VIE tags can be successfully applied to recently hatched tadpoles, (2) tags can be reliably followed throughout larval development and sometimes retained across metamorphosis, and (3) VIE tags do not appear to interfere with parental care behavior (i.e., tadpole transport). Our study expands the application of tagging to early developmental stages in tropical amphibians which can be of use in behavior, conservation, and natural history research studies in the future.

## ACKNOWLEDGEMENTS

We would like to thank German Orizaola, Andis Arietta, Max Lambert, Janne Valkonen, Lutz Fromhage, and one anonymous reviewer for such constructive and supportive comments. We also owe an enormous thank you to Matthieu Bruneaux for his conversation and recommendation with respect to Bayesian modeling. Finally, *kiitos* to our master's students Nina Kumpulainen and Emmi Alanen, and lab technician Teemu Tuomaala for helping with tadpole care—it takes a village.

### Funding

This study was funded by the Academy of Finland (Academy Research Fellowship No. 21000042021 to Bibiana Rojas). The funders had no role in study design, data collection and analysis, decision to publish, or preparation of the manuscript.

### Grant Disclosures

The following grant information was disclosed by the authors:
Academy of Finland: 21000042021.

### Competing Interests

The authors declare that they have no competing interests.

### Author Contributions

- Chloe A. Fouilloux conceived and designed the experiments, performed the experiments, analyzed the data, prepared figures and/or tables, authored or reviewed drafts of the paper, and approved the final draft.
- Guillermo Garcia-Costoya conceived and designed the experiments, performed the experiments, analyzed the data, prepared figures and/or tables, authored or reviewed drafts of the paper, and approved the final draft.

- Bibiana Rojas conceived and designed the experiments, authored or reviewed drafts of the paper, and approved the final draft.

## Animal Ethics

The following information was supplied relating to ethical approvals (i.e., approving body and any reference numbers):

The National Animal Experiment Board approved this research (ESAVI/9114/04.10.07/2014).

## Data Availability

The raw measurements, Bayesian analysis upon which the models are constructed, and the growth and survival code are available as Supplemental Files.

Data and code are available at the JYU data repository: https://jyx.jyu.fi/handle/123456789/71219.

## Supplemental Information

Supplemental information for this article can be found online at http://dx.doi.org/10.7717/peerj.9630#supplemental-information.

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
