# Peer review of "Visible implant elastomer (VIE) success in early larval stages of a tropical amphibian species"

_PeerJ, doi:10.7717/peerj.9630_

## Round 0.1 · original submission · Minor Revisions

Your manuscript has now been reviewed by three reviewers and myself. In general, all reviewers enjoyed your manuscript and found the research to be well done. Each reviewer has identified a number of helpful edits for you. In particular, I call you attention to a number of issues relating to 1) restructuring and focusing your introduction and discussion a bit more and 2) adding substantially more detail to your methods to make this technique easier to follow for your readers and to better interpret your results.

I have a few additional comments as well:

Line 63: Marking is not only useful for behavior but for population biology in general.

Lines 136-138: Given the importance of the anesthetic here, I’d be keen to see the data for the statement that the potency did not degrade after repeated usage. That seems like it would be valuable.

Lines 240-252: I’d suggest maybe using a binomial mixed effects model to look at whether initial weight or Gosner Stage influenced the probability of tagging success.

Anesthesia and Tadpole tagging: I think we need more detail on the timing and order of operations for removing tadpoles from anesthesia, drying them, injecting them, recovering them in fresh water if that happened, etc.

Line 283: Please see Warne and Crespi (2015; Larval growth rate and sex determine resource allocation and stress responsiveness across life stages in juvenile frogs). This study marked larval wood frogs (though not as early in development as you did) and carried the mark through metamorphosis for over ¾ of the larvae.

Congratulations on your well-done study and manuscript; I look forward to seeing your revision.

·

Basic reporting

No comment.

Experimental design

The study is well designed. However, it would have been much more interesting, and probably more similar to field conditions, if it would have included more than one VIE color and different sides of the tadpoles, simulating multi-marking in the field.

Validity of the findings

The main novelty of the study relays in the application of a well-known technique (VIE) to a new group of organisms, at a particularly small size. This on itself, is not a great novelty, but opens the field for conducting much more detailed studies on the ecology and behaviour of Dendrobatid frogs, which present many characteristics that make them especially interesting study subjects.

Additional comments

This is a nice methodological study that presents the results of VIE marking in Dendrobatids all across the larval development. Authors were able to mark individuals right after hatching and followed them until metamorphosis. The study reports the VIE marking of the smallest tadpole until now. The study presents a high rate of VIE retention that will allow to export this technique to field conditions in order to conduct detailed studies on the ecology and behaviour of these species.

Overall, this is an interesting study, with lot of potential to influence future work. I have only minor comments that will hopefully improve the manuscript.

Line 45: The range of marking techniques included here is a bit limited. I guess it should be good to include some of the more used methods for individual marking in wildlife e.g. metal and colour rings (birds), ear tags or collars (mammals)...

Line 48. Substitute "frogs" by "amphibians" in order to be a bit more general.

Line 63. Expand this comment. Individual identification is important not only for behavioural studies, but for studies on physiology, life-history, and ecology in general.

Lines 64-65. Expand this sentence also, considering the previous comment. Add more examples here.

Lines 67-68. As mentioned before, much more examples of widespread techniques including metal and colour rings in birds, ear tags and collars in mammals, GPS/radiotracking devices.

Line 80. Change “anurans” by “amphibians”, more general.

Line 86-89. Tadpoles are not especially plastic in the tropics. They are, overall, highly plastic in temperate regions too as many studies have shown (see studies by Gómez-Mestre, Laurila, Merilä…).

Line 92. I suggest to add a few examples about the interest of individual marks in amphibians, before moving to the specific case of poison frogs.

Lines 97-99. These lines are the classical ones you expect to find on the last paragraph of the Introduction. I short of understand the rationale of having something like this here. I suggest authors try to find a good way to move them from here to the end of the Intro, and simply introduce here the study species, or something similar.

Line 99. Change “is” by “are”.

Methods. I miss a more general description of what VIE are. For the ones we are familiar with VIE elastomers it is quite clear, but more general audiences would benefit for a more detailed description of the material, different types of elastomers (colours, possibility of UV detection…).

As a general methodological comment. Since the idea is to test this method for future use in the field, it would have been better to test combinations of colours, and sides of the tadpoles in order to generate a larger set of codes. Understanding the detectability and relocation of colour combinations is key for the use of this technique in the field.

Statistics. I am not proficient on Bayesian models, so I cannot assess this section properly.

Line 250. I think authors should be careful when commenting on the role of the methods for transport behaviour studies due to the really small sample size (n= 2). I suggest to say that the study “suggests” that VIE do not prevent, rather than using more absolute terms (“did not prevent”). Same for lines 288-289.

Line 301-302. Mortality during early developmental stages in amphibians is high, no doubt, but I guess here is probably due to sub-optimal conditions in the lab (feeding, probably) more than anything.

Lines 316-324. These results on the relocation of VIE are really interesting.

Figure 1. Not sure about this formula as figure 1.

Figure 4. “growth rate percent” is a rather unusual way to express growth rate in tadpoles. I suggest to change this trait to mg/day or something similar, more often used in other studies.

·

Basic reporting

The authors report a study assessing a new anesthetization technique coupled with a trial of VIE label marking in early stage Dendrobates tinctorius.

As the authors point out, marking methods have been disproportionately focused on temperate species; so, it is great to see a study on a tropical species. This study is impressive in the small size of tadpoles that were marked. Methods for marking early stages are sorely needed but few methods exist. I applaud the author’s success at marking such small tadpoles (and for attempting to even mark embryos).

Overall, I found the writing easy to follow. Although I have some questions and suggestions about the methods/analysis, I thought the experimental design and their interpretation of the results were straight forward.

There are two main issues I have with the manuscript as written. First, the observational methods need to be detailed much more explicitly since the main results hinge on them. Second, the issue of marking through metamorphosis needs to be addressed formally. I detail these in subsequent sections.

I had a lot of fun reading this manuscript and appreciate the hard work that the authors put into this study.

Experimental design

In general, I think the experimental design is reasonable. However, I do have some important concerns.

My main criticism of the methods is that the main result (successful mark retention) is contingent on observation of the tag; however, there is not enough description of the observation method. For instance, did a single person observe at a time or did multiple people look for the mark to validate? Did the same person observe the larvae at each every time-point? Without knowing the answer, it is hard to know how confident we should be in a false negative—were false negatives the result of observer error or the mark physically migrating during development?

It is similarly difficult to know how confident we should be about positive observations. For instance, were observations conducted blind to control for bias? Were control animals also assessed? If so, it would be useful to report the false positive rate (e.g. in Supplemental Figure 5 it looks like there are other fluorescent particles that might lead to false positives).

Validity of the findings

First, I want to thank the authors for including very easy-to-follow code. I especially appreciated that they left their comments in the code, which made it easy to understand their justifications.

Overall, I think the stats are sensible, especially given their sample sizes. However, I think the authors need to provide more justification for why they are mixing inference frameworks. For instance, tag retention and survival could have been modeled with frequentist mixed models in the same way that growth rates were analyzed. Or, Bayesian methods could have been employed for all. I fully trust that the authors had good reasons for the analysis strategies they chose, but would like to know why.

A few specific points:

I would like to see model report tables for growth rates and survival somewhere (supplemental information is fine). This information is included for the mark retention analysis but not for the other models.

One minor, but potentially important issue in the statistics code is that, when comparing linear mixed models, model should be fit with maximum likelihood. The authors are comparing models fit with restricted maximum likelihood, which is invalid (see Zuur's 2009 book on mixed effect models). Also, note that there is no link function specified in the glm model, which means that it is fitting the same model as the lm model (this is apparent when comparing AIC values). I seriously doubt that this will influence the results, so I'm not too worried, but just a good practice to be aware of.

Throughout the manuscript, the authors mention the success of their technique for marking across metamorphosis. I realize that mortality precluded formal analysis through metamorphosis, but some reporting on mark retention in metamorphs needs to be included. Given the limited number of metamorphs, I'm not sure strong claims about marking success between lifestages are warranted.

Additional comments

Here are some comments about specific sections or lines in the text:

The introduction seems broad for the context of this study. I understand that the intent is to make the case that marking is useful in biological studies, but perhaps it would be better to anchor the motivation for this study more in amphibians rather than large mammals? There are lots of reasons herp folks need to mark individuals that are directly relevant.

Line 78-79: “long-term marker-based studies have not often been applied to animals with complex life cycles” is not necessarily true. Marked have been used on animals with complex life cycles, but only in adult stages. I think you mean that marking “across ontogenetic stages” has not been common.

Line 85-89: It sounds like you are making the argument that tagging success may differ because neotropical species are more plastic than temperate species, which, I don’t think is true, at least morphologically and physiologically.

Line 97-99: The last sentence seems out of place in this paragraph. Maybe it would fit better in the final paragraph of the introduction?

Line 103: Just a note: Andis (2018) tagged larvae earlier than stage 30 to assess mark retention within the larval stage, but only considered inter-stage marking through metamorphosis in larvae over stage 30.

Line 111: You cite Gosner (1960) here but the first reference to Gosner staging is in the prior paragraph.

Given the focus on the somewhat unique natural history of D. tinctorius (e.g. parental care, larval transport, etc.), it might be useful to include some natural history for the species in the “Study organism” section. At the very least, I would like to know how long the typical larval period is for this species in the wild. Also, since other larval marking techniques have been implicated in reduced fitness and increased predation, it would be good to know what relevant selection pressures D. tinc. larvae face in nature.

Lines 122-132: This paragraph switches between past and present tense quite a bit.

Line 163-164: Out of curiosity, does this mean that the elastomer is waning the whole time or is there some threshold after which it starts to diminish?

Line 184-188: This does not seem to make the result very applicable to field use since, in the wild, an undetected mark will be categorized as “unmarked” and would not be associated with prior or subsequent observations (since, in the field, individual ID is reliant on the mark, unlike the lab where individual ID is known regardless). Maybe a better way to present this information would be to report false negative rates (and false positive rates, if you can) over time. This would have the added benefit of illuminating any ontogenetic pattern (if, for instance, there is some stage in which integumentary development tends to mask the mark).

Line 240-262: Somewhere in the results the number of metamorphs and detection rate for metamorphs needs to be reported. (I recognize that mortality made it impossible to make statistical inferences about post-metamorphic retention, but the raw numbers can still be reported). It would also be useful to know at which week larvae tend to metamorph, since all the graphs and models deal with time in weeks rather than developmental stage.

Line 373-377: It might be worth discussing how fluorescent marks may impact survival in the wild. For instance Carlson and Langkilde (2015) found that a staining procedure made larvae more susceptible to predation. Given that amphibians have evolved specific color patterns for a reason, it seems reasonable to consider that altering that pattern very well may inhibit crypsis or aposematic signaling.

Reviewer 3 ·

Basic reporting

The manuscript is clearly written and easy to read. The introduction and background are well-supported with references to the literature. The manuscript conforms to PeerJ’s structure, the figures are well-presented (although I think one may be unnecessary – see minor comments), and the raw data are supplied (see raw data check).

Experimental design

This is an original experimental study with a well-defined research question. There are a few aspects of the study and methods which I think need additional information (see comments) to be replicable but the vast majority of the information is present.

Validity of the findings

Conclusions are generally well stated and clearly supported.

Additional comments

The manuscript entitled “Visible implant elastomer (VIE) success in early larval stages of a tropical amphibian species” presents an experimental lab study. The authors marked larval frogs at an early developmental stage and tracked them across development to examine the persistence of the marking and to understand any effects of early marking on growth and survival by comparing with unmarked individuals. The authors found that marks persisted at a remarkably high rate for the first month with declining retention over time. However, some individuals retained their marks through metamorphosis. In addition, the authors found no effect on growth or survival as a result of the marking procedure. The study is well-written and the motivation is well-explained. This manuscript will appeal strongly to anyone interested in marking animals at early life stages, particularly researchers studying organisms with complex life histories. In addition, I thought the authors’ discussion of mark retention vs mark detection was very thoughtfully done and I’m glad they included it. Below, I detail my comments for the submitted manuscript.

-Persistence of marks and study of the animals over time:

Much of your introduction is framed around the idea that researchers are in need of a marking method that can be applied to an organism at a very early developmental stage and a small size that will persist through metamorphosis. I agree with this framing and feel your study can provide an important tool here. However, there were some important details missing from the manuscript that prevented me from feeling that I fully understood how well the marks persisted over time and how the animals were kept during that time period.

How long were the animals housed in the manner described at the end of the first paragraph of the methods? Your figures give data across a 12-week time span but it’s not written in the methods or results. In addition, Line 243 says that 3 months is approximately the amount of time that these animals took to reach metamorphosis and Line 283 says that this is the only study to follow individuals across metamorphosis but no mention is made of how the animals would be housed after metamorphosis. How long is time to metamorphosis, on average, for this species? Did you house post-metamorphic animals for the duration of this study? If so, how were they kept?

Did you follow all individuals all the way through metamorphosis? You indicate in several places that there was high mortality around metamorphosis and high juvenile mortality but how many individuals survived to metamorphosis? Past metamorphosis? And for how long? Much of the follow through to your introduction depends on your reader understanding that the marks can persist across metamorphosis. However, apart from telling us that they did so in the discussion, this information is not provided in the methods or results.

The raw data seems to indicate that animals were marked in October 2019 and some animals were followed through April 2020. Given that I am reading your manuscript in May 2020, it is completely reasonable that you stopped analyzing these animals at some point for this manuscript. However, please explicitly tell us the time span that the analyses here covered and the number of individuals surviving at the end of that time.

-Novel anesthetic:

You mention in the abstract and in the introduction that you are using a novel anesthetic, 2-PHE. I think it would be useful for your readers to give some additional information about it. You give us a brief explanation at the end of the introduction, lines 113-115 – that it does not need to be buffered and can be stored at room temperature which makes it advantageous for field work. This is helpful but I’d also like a bit of background. Has it been in use as an anesthetic previously in other organisms? What were its previous applications? Can it be handled safely and easily? Is it difficult to obtained?

For non-amphibian researchers, it might be useful to talk briefly about how amphibians are usually anesthetized and why you think 2-PHE is advantageous.

Also, on Line 136, tell us more specifically how many times the solution was used within a day? It doesn’t need to be exact as I can guess that the number of individuals that you tagged on given day varied. However, an idea (5, 10, 20?) would be useful to give the reader an understand of how many times the anesthetic can be used within a day, once prepared.

Minor comments:

-The reference list is duplicated within the Reference section. The repeat begins on Line 545.

-Figure 1 was unnecessary for me. I thought your verbal description of the statistical methods was very thorough and I understood how tag retention was marked in your dataset without it so I think you could remove it or move it to the supplement.

---

## Round 0.2 · accepted · Accept

Thank you for your thorough responses to the reviewers and revisions. The manuscript is greatly improved in content and clarity. Congratulations on an excellent piece of work.